# Factors associated with pentavalent vaccine coverage among 12-23-month-old children in Afghanistan: A cross-sectional study

Muhammad Kamel Frozanfar[1,2], Nobuyuki Hamajima[1], Said Hafizullah Fayaz[1,3], Abdullah Darman Rahimzad[4], Hedayatullah Stanekzai[5], Souphalak Inthaphatha[1], Kimihiro Nishino[1], Eiko Yamamoto[1] *

1 Department of Healthcare Administration, Nagoya University Graduate School of Medicine, Nagoya, Aichi, Japan, 2 Healthcare-Associated Infections and Antimicrobial Resistance Program, Office of Epidemiology, Virginia Department of Health, Richmond, Virginia, United States of America, 3 National Heart, Lung and Blood Institute, National Institute of Health, Bethesda, Maryland, United States of America, 4 Department of Ear, Nose and Throat, Balkh University Faculty of Medicine, Mazar-i-Sharif, Balkh, Afghanistan, 5 National Emergency Operation Center, Ministry of Public Health, Kabul, Afghanistan

* yamaeiko@med.nagoya-u.ac.jp

**Data Availability Statement:** Data cannot be shared publicly because of a property of the Ministry of Public Health (MoPH) of Afghanistan.

## Abstract

### Introduction

This study aimed to identify the factors associated with the coverage of the third dose of pentavalent vaccine (Penta3) among children aged 12–23 months in Afghanistan.

### Methods

The data of 3,040 children aged 12–23 months were taken from the Afghanistan Health Survey 2018, including characteristics of the children and their households, household heads, and mothers/primary care givers. Adjusted odds ratios (AORs) and 95% confidence intervals (CIs) were estimated using a logistic model. Multivariable stepwise logistic regression analysis with forward-selection (Model 1) and backward-selection (Model 2) was performed using variables that showed significant differences by bivariate analysis.

### Results

The coverage of Penta3 among 12–23-month-old children was 82.3%. Factors associated with Penta3 coverage in the two models of multivariable analysis were 18–23 months old compared to 12–17 months old; having no diarrhea in the last two weeks compared to having diarrhea; no bipedal edema compared to having edema; taking vitamin A supplement; 1–2 children under five years in a household compared to three or more; distance from residence to the nearest health facility ≤2 hours on foot; having a radio; having a TV; educated heads of households; non-smoking of heads of households; and literacy of mothers/primary caregivers.

MoPH of Afghanistan has the ownership and authority to manage the AHS 2018 data. Only authorized and permitted researchers by MoPH are allowed to access and use the data for research and analytical purposes. Data are available from the Ministry of Public Health of Afghanistan (General Directorate of Monitoring & Evaluation and Health Information System; Tel +93-20-231-3688 | Emails ehis.moph@gmail.com, mehis.moph@gmail.com) for researchers who meet the criteria for access to confidential data.

**Funding:** The author received no specific funding for this work.

**Competing interests:** The authors have declared that no competing interests exist.

## Conclusions

Penta3 coverage among 12–23-month-old children improved but was still lower than the target. Primary education should be provided to all children throughout the country. TV and radio are useful tools for providing health information. Mobile outreach programs and the establishment of new health facilities should be promoted to improve access to health service for all people in Afghanistan.

## Introduction

Childhood immunization can protect children from vaccine-preventable diseases and it is considered as the most cost-effective healthcare investment that has long-term benefits for public health [1–3]. According to the estimations, immunization annually averts 2.5 million child deaths around the world from vaccine-preventable diseases [2, 4]. The World Health Organization (WHO) established the Expanded Program on Immunization (EPI) in 1974 for extending immunization activities worldwide. Initially the target of this program was six diseases: tetanus, diphtheria, poliomyelitis, pertussis, tuberculosis, and measles [5]. It is important that different vaccine types are administered in a single dose to avoid multiple shots to the target children. Fortunately, the introduction of combined vaccines has promoted the incorporation of additional vaccines into a vaccination schedule. In 2011, the pentavalent vaccine (Penta), which protects against diphtheria, pertussis, tetanus, hepatitis type B, and hemophilia influenza type B, was introduced in many countries by the Global Alliance for Vaccines and Immunizations [6, 7].

Afghanistan is one of the countries that has the highest mortality rate of children under five years old (under-five mortality) in the world, although the under-five mortality in Afghanistan declined from 191 per 1,000 live births in 2006 to 58 per 1,000 live births in 2020 [8, 9]. One of the reasons for the high under-five mortality rate is low vaccination coverage. In Afghanistan, the Ministry of Public Health (MoPH) started the national immunization program in 1978 [10] and eight vaccines are provided to prevent vaccine-preventable diseases, such as Bacillus Calmette–Guerin (BCG), measles, oral polio vaccine (OPV), and Penta. According to the current EPI schedule, BCG and OPV 0 dose (OPV0) are administered at birth, Penta1 and OPV1 in the 6th week, Penta2 and OPV2 in the 10th week, Penta3 and OPV3 in the 14th week, measles first dose (MCV1) in the 9th month, and MCV2 in the 18th month. All vaccines except MCV2 should be administered to all children before they are one year old [2, 5, 8]. The WHO Eastern Mediterranean Region set the target of childhood vaccine coverage for all antigens as 90% for countries in the region, including Afghanistan [11]. However, the coverage of Penta3 was 58% in 2015, and there is still much to do in order to reach the optimal 90% coverage countrywide [12].

Since the EPI started in Afghanistan, the vaccine delivery system has been improving. EPI centers, which have at least one trained vaccinator, provide the routine immunizations free of charge and the number of EPI centers increased from 870 in 2004 to more than 1,767 in 2016 [13]. The routine immunization coverage improved from 30.1% in 2003 to 59.7% in 2013 [10], but decreased from 58.8% in 2015 and 50.2% in 2018 [8, 14]. The coverage in 2018 was low (3.1–28.1%) in southern and southeastern provinces. Therefore, Afghanistan still has strong outbreaks of vaccine-preventable diseases, especially diseases that can be prevented by Penta. In 2003, providing maternal and child health services by mobile health teams, which consist of a midwife, a vaccinator, and a nurse, started in remote and conflict-affected areas. In 2016 and

2017, the coverage of Penta3 in the areas with mobile health team services was 26.6%, which was lower than that in the areas without the services (30.5%) [15]. To improve the coverage of Penta3, the factors affecting no or incomplete vaccination of Penta3 should be understood. Previous studies reported that the factors associated with non- and under-vaccination of children in Afghanistan were socio-demographic and childhood factors, lack of urbanization, access to health facilities, and household factors including maternal and paternal education [2, 5, 12]. However, to our knowledge, there have been no studies on the factors associated with Penta3 coverage in Afghanistan. Therefore, this study aimed to identify factors associated with Penta3 coverage among children aged 12–23 months in the country, using the data of the Afghanistan Health Survey (AHS) 2018.

## Materials and methods

### The AHS 2018

This study is a secondary analysis of the AHS 2018 data. The AHS 2018 was conducted by the MoPH and Koninklijk Instituut voor de Tropen in 2018, covering all 34 provinces of Afghanistan [14]. The AHS 2018 was approved by the Institutional Review Board of the MoPH in Afghanistan. In this survey, stratified two-stage cluster sampling was used. Stratification was achieved by separating each province into urban and rural areas. In the first stage, 30 clusters were sampled per province. In the second stage, 23 households were sampled for household interviews in each selected cluster, yielding 690 households per province. Therefore, 23,460 households (4,738 households in urban areas and 18,722 households in rural areas) were selected as the survey samples in all 34 provinces. However, less than two-thirds of the selected clusters were surveyed in Zabul, Nuristan, and Helmand Provinces due to civil wars and hard-to-reach areas [14]. A total of 19,684 households from 912 clusters in 34 provinces were finally included in the survey.

Data collection was conducted from March to August in 2018 [14]. Field teams consisted of male and female interviewers, recruited from each of the 34 provinces of Afghanistan. All staff received extensive training prior to going into the field. The AHS instruments consisted of six questionnaires: (1) general information and socio-demography, (2) household history, (3) anthropometry of children, (4) mother's general information, (5) women's pregnancy history, and (6) child immunization status. The field teams visited each selected household and interviewed all men who had ever been married and women aged 15–49 of the household using the questionnaires after obtaining verbal consent. The response rate of households and women was 94.4% and 93.9%, respectively [14]. Regarding children's data collection, the anthropometric completion rate and child health completion rate was 93.8% and 94.3%, respectively. Data collection in the field took place by adequate interview techniques maintaining all ethical protocols. When the data were analyzed, sampling weights were applied in order to adjust the results for differences that might be caused by the selection steps.

### Study participants

In this study, the study subjects were children aged 12–23 months old, their mothers/primary care givers, and their household heads who participated in the AHS. Of the 19,684 households included in the AHS 2018, 3,040 households had a child aged 12–23 months and there was no household that had two or more children aged 12–23 months. Information of children was obtained from the mothers or primary caregivers of the target children. Heads of households and mothers/primary caregivers of 3,040 children agreed to participate in the survey including children's information. The characteristics of the 12–23-month-old children, the

characteristics of their households, and the characteristics of their household-heads and mothers/caregivers were taken from the AHS dataset. The dataset was provided by the MoPH for this study.

## Variables of children

Characteristics of the children included sex, age, having a vaccine record card, having Penta1 and Penta3, diarrhea in the last two weeks and at the interview, bipedal edema at the interview, and taking vitamin A supplement. Age was categorized into 12–17 months old and 18–23 months old. History of Penta was collected from their vaccination cards or recalls of their mothers/primary caregivers. The definition of diarrhea was the passage of three or more loose or liquid stools in 24 hours [16]. Mothers/primary caregivers were asked about vitamin A supplementation for their children during the last national immunization campaign. In Afghanistan, vaccine record cards do not record vitamin A supplementation. Children were inspected by the investigation team if they had bipedal edema, which is an indicator of acute malnutrition [17].

## Variables of households

Socio-demographic characteristics of the households were location, number of total people, number of total eligible women, number of total children under five years old, type of residence, source of drinking water, type of toilet, distance from the house to the nearest health facility, and having a radio, a TV, and a mobile phone. A household was defined as a group of people who lived in the same house and were led by the same household head. Location of a household was determined as urban or rural based on the criteria by the government. The total number of people in a household was categorized into 1–5, 6–10, and 11 or more. 'Eligible women' means women who were of reproductive age (12–49 years old). The number of eligible women (1 or 2≤) and children under five years old (1–2 or 3≤) in a household were categorized into two groups considering nuclear or extended families. Type of residence was designated as their own house or a rental house. The sources of drinking water were categorized into protected sources (covered wells, hand pumps, pipes, and public tap) and unprotected sources (open well, water from spring, rainwater, surface water, and ponds). Toilets were classified into sanitary and unsanitary. Sanitary toilets hygienically prevent contact with human excreta, which included flush or pour-flush toilets, piped sewer systems, septic tanks or pits, ventilated improved pit latrines, pit latrines with slabs, and composting toilets. Distance from a house to the nearest health facility was categorized into less than 30 minutes, 30 minutes to two hours, and more than two hours on foot. The data collectors observed whether a household had a TV, radio, and mobile phone.

## Variables of household heads and mothers/primary caregivers

Characteristics of the heads of household included sex, age, education, employment type, marital status, smoking cigarettes, and usage of narcotic drugs. A head of the household was defined as a person in the household who made major decisions on financial expenditures, medical care, schooling, and food supply of the family. Age groups were categorized considering the quantile of the age (≤29 years, 30–44 years, 45–59 years, and ≥60 years). The level of education was grouped into "yes" when heads of households had primary or higher education. Employment type was grouped as self-employed or paid worker (person who worked for the government or private sector). Marital status was categorized into "married" and others (single, divorced, widow, or widower). They were asked about if they smoked cigarettes and if taking narcotic drugs.

Characteristics of mothers/primary caregivers included age, literacy, watching TV, and listening to the radio. Age was categorized into four groups (≤17 years, 18–29 years, 39–44 years, and ≥45 years). Literacy of mothers/primary caregivers were categorized after observing their reading and writing ability of any of the common languages of the country (Pashto, Dari, or Uzbek language). The women were asked how often they watched TV or listened to the radio and the response of "not at all" was categorized into "no" and the others (almost every day, at least once a week, and less than once a week) were categorized into "yes." Variables for this study were chosen according to previous studies on the vaccine coverage [2, 5, 12, 18–21].

## Statistical analysis

Descriptive analysis was used to describe the characteristics of children, their households, and their household heads and mothers/caregivers. A logistic regression analysis was applied to obtain odds ratio (OR) and 95% confidence interval (CI). Multivariable stepwise logistic regression analysis was performed on Penta3 using 18 variables that showed significant differences by bivariate analysis (Model 1, forward-selection; Model 2, backward-selection). Multicollinearity was not found between the variables according to the variance inflation factor values for each of variables. The associations were considered statistically significant when $P$-value was <0.05. Statistical Package for the Social Sciences version 24.0 (IBM SPSS Inc, Armonk, NY, USA) was used for analysis.

## Ethical issues

In this study, written informed consent was waived due to a secondary analysis of the anonymous data of the AHS 2018 and the study protocol was approved by the Institutional Review Board of the MoPH in Afghanistan (Approval number: E.1019.0087). In the AHS, field processes included appropriate interview techniques that incorporated all relevant ethical protocols. Verbal consent was obtained from each participant, namely a household head and a mother/primary caregiver of each child, after interviewers read all contents of a verbal consent form to the participant. For data collection of a child, verbal consent was obtained from the mother/primary caregiver of the child. When verbal consent was obtained, the interviewers checked the box of "consented", wrote their name and the date of interview, and signed on the consent form.

## Results

The data of 3,040 children aged 12–23 months, their mothers/primary caregivers, and heads of their households were taken from the AHS 2018 dataset. Of the 3,040 children, 1,666 children (54.8%) were male and 1,742 children (57.3%) were 12–17 months old (Table 1). The average age was 16.5 months old. Most children (n = 2,646, 87.0%) had vaccination cards but 340 children (11.2%) did not have vaccination cards.

The coverage of Penta1 and Penta3 was 94.0% (n = 2,857) and 82.3% (n = 2,502), respectively. Children who had a history of diarrhea in the last two weeks accounted for 32.7% (n = 997) and 146 children (4.8%) had diarrhea at the time of data collection. Bipedal edema was found among 44 children (1.4%) and 2,602 children (85.6%) took vitamin A supplements when they had the last vaccination.

Most households lived in rural areas (n = 2,442, 80.4%) and comprised of 6–10 persons (n = 1,338, 44.0%) and one eligible woman (n = 1,612, 53.0%) (Table 2). Households with one or two children under five years accounted for 66.4% (n = 2,019). The average number of total people, eligible women, and children under five years old in a household was 10.9, 1.7, and 2.4, respectively. Most households lived in their own residence (n = 2,579, 84.8%), had protected

**Table 1. Characteristics of 12–23-month-old children (N = 3,040).**

| Variables | | Total (N = 3040) | Penta3 (N = 2502) | |
|---|---|---|---|---|
| | | N (%) | n | % (95% CI) |
| Sex | | | | |
| | Female | 1374 (45.2) | 1120 | 81.5 (77.2–85.8) |
| | Male | 1666 (54.8) | 1382 | 83.0 (79.0–87.0) |
| Age (months old) | | | | |
| | 12–17 | 1742 (57.3) | 1469 | 84.3 (80.3–88.3) |
| | 18–23 | 1298 (42.8) | 1033 | 79.6 (75.3–83.9) |
| Vaccination recorded on the card | | | | |
| | Yes | 2646 (87.0) | 2190 | 82.8 (79.6–86.0) |
| | No | 340 (11.2) | 268 | 78.8 (70.4–87.2) |
| | No response | 54 (1.8) | 44 | 81.5 (59.8–103.2) |
| Penta1 | | | | |
| | Yes | 2857 (94.0) | 2502 | 87.6 (84.4–90.8) |
| | No | 183 (6.0) | 0 (0.0) | 0.0 (0.0–0.0) |
| Diarrhea in the last 2 weeks | | | | |
| | No | 2043 (67.3) | 1712 | 83.8 (80.2–87.4) |
| | Yes | 997 (32.7) | 790 | 79.2 (74.3–84.1) |
| Diarrhea at the time of the interview | | | | |
| | No | 2894 (95.2) | 2379 | 82.2 (79.2–85.2) |
| | Yes | 146 (4.8) | 123 | 84.2 (70.5–97.9) |
| Bipedal edema | | | | |
| | Absent | 2996 (98.6) | 2472 | 82.5 (79.5–85.5) |
| | Present | 44 (1.4) | 30 | 68.2 (48.0–88.4) |
| Vitamin A supplementation | | | | |
| | No | 438 (14.4) | 325 | 74.2 (67.3–81.1) |
| | Yes | 2602 (85.6) | 2177 | 83.7 (80.5–86.9) |

Penta1, the first dose of pentavalent vaccination; Penta3, the third dose of pentavalent vaccination.

water sources (n = 2,244, 73.8%) and unsanitary toilets (n = 2,344, 77.1%), and located at a distance of 30 minutes up to 2 hours on foot from the nearest health facility (n = 2,790, 91.8%). Of the 3,040 households, 83.2% (n = 2,528) had mobile phones but only 36.7% (n = 1,115) and 46.3% (n = 1,406) had radios and televisions, respectively.

Most heads of households were male (n = 3,000, 98.6%) and 30–44 years (n = 1,117, 36.7%) and the average age was 43.6 years old (Table 3). There were 8 (0.3%) and 40 (1.3%) household heads who were under 18 and 21 years, respectively. Only 39.5% of the household heads (n = 1,201) had primary or higher education and 52.2% (n = 1,588) were paid workers. Only 9.4% and 2.6% of heads of households were smokers and users of narcotic drugs, respectively.

The average age of 3,040 mothers/primary caregivers was 28.9 years old and 60.6% of all mothers were 18–29 years old. Mothers/primary caregivers who could read any of the local languages accounted for 24.1% (n = 732) (Table 3). Only 27.7% (n = 841) and 39.4% (n = 1,199) of all mothers/primary caregivers answered that they listened to the radio and watched TV, respectively.

The number and percentage of children who had Penta3 in each category are shown in Tables 1–3. Binary logistic regression analysis showed that children who were 18–23 months old (OR = 0.72, 95%CI 0.60–0.87), who had diarrhea in the last two weeks (OR = 0.71, 95%CI 0.60–0.89), and having bipedal edema (OR = 0.45, 95%CI 0.24–0.86) were negatively

**Table 2. Socio-demographic characteristics of the households (N = 3,040).**

| Variables | | Total (N = 3040) | Penta3 (N = 2502) | |
|---|---|---|---|---|
| | | N (%) | n | % (95% CI) |
| Location | | | | |
| | Rural | 2442 (80.4) | 1971 | 80.7 (77.5–83.9) |
| | Urban | 598 (19.6) | 531 | 88.8 (81.7–95.9) |
| Total people | | | | |
| | 1–5 | 486 (16.0) | 411 | 84.6 (77.1–92.1) |
| | 6–10 | 1338 (44.0) | 1134 | 84.8 (80.3–89.3) |
| | 11≤ | 1216 (40.0) | 957 | 78.7 (74.3–83.1) |
| Total eligible women | | | | |
| | 1 | 1612 (53.0) | 1360 | 84.3 (80.2–88.4) |
| | 2≤ | 1428 (47.0) | 1142 | 79.9 (75.8–84.0) |
| Total children under five years old | | | | |
| | 1–2 | 2019 (66.4) | 1716 | 85.0 (81.3–88.7) |
| | 3≤ | 1021 (33.6) | 786 | 77.0 (72.3–81.7) |
| Residence | | | | |
| | Own | 2579 (84.8) | 2110 | 81.8 (78.6–85.0) |
| | Rental | 461 (15.2) | 392 | 85.0 (77.2–92.8) |
| Source of drinking water | | | | |
| | Protected | 2244 (73.8) | 1835 | 81.8 (78.4–85.2) |
| | Unprotected | 796 (26.2) | 667 | 83.8 (78.0–89.6) |
| Type of toilet | | | | |
| | Sanitary | 696 (22.9) | 598 | 85.9 (79.5–92.3) |
| | Unsanitary | 2344 (77.1) | 1904 | 81.2 (77.9–84.5) |
| Distance to the nearest health facility (on foot) | | | | |
| | <30 min | 194 (6.4) | 161 | 82.9 (71.2–94.6) |
| | 30 min–2 hr | 2790 (91.8) | 2326 | 83.4 (80.3–86.5) |
| | 2 hr< | 56 (1.8) | 15 | 26.8 (19.8–33.8) |
| Having a radio | | | | |
| | No | 1925 (63.3) | 1612 | 83.7 (80.0–87.4) |
| | Yes | 1115 (36.7) | 890 | 79.8 (75.1–84.5) |
| Having a television | | | | |
| | No | 1634 (53.7) | 1307 | 80.0 (76.1–83.9) |
| | Yes | 1406 (46.3) | 1195 | 84.9 (80.5–89.3) |
| Having a mobile phone | | | | |
| | No | 512 (16.8) | 399 | 77.9 (71.2–84.6) |
| | Yes | 2528 (83.2) | 2103 | 83.2 (80.0–86.4) |

Penta3, the third dose of pentavalent vaccination.

associated with Penta3 (Table 4). Children who took a vitamin A supplement when having the last vaccine (OR = 1.80, 95%CI 1.40–2.26), who lived in urban areas (OR = 1.89, 95%CI 1.44–2.49) and whose households had one eligible woman compared to two or more eligible women (OR = 2.76, 95%CI 2.07–3.68), had 1–2 children compared to three children or more (OR = 1.69, 95%CI 1.40–2.05), and had sanitary toilets (OR = 1.41, 95%CI 1.11–1.78) had significantly more Penta3. Children who lived at a distance of more than two hours on foot from the nearest health facility (OR = 0.07, 95%CI 0.04–0.13) were negatively associated with Penta3. Children whose household had a radio (OR = 1.30, 95%CI 1.07–1.57), had a TV

**Table 3. Characteristics of the household heads and mothers/primary caregivers (N = 3,040).**

| Variables | | Total (N = 3040) | Penta3 (N = 2502) | |
|---|---|---|---|---|
| | | N (%) | n | % (95% CI) |
| Sex of household head | | | | |
| | Male | 3000 (98.6) | 2469 | 82.3 (79.4–85.2) |
| | Female | 40 (1.4) | 33 | 82.5 (56.9–108.1) |
| Age of household head (years old) | | | | |
| | ≤29 | 736 (24.2) | 621 | 84.3 (78.2–90.4) |
| | 30–44 | 1117 (36.7) | 931 | 83.3 (78.4–90.4) |
| | 45–59 | 774 (25.5) | 612 | 79.1 (73.5–84.7) |
| | 60≤ | 413 (13.6) | 338 | 81.8 (73.9–89.7) |
| Education of household head | | | | |
| | No | 1839 (60.5) | 1474 | 80.2 (76.5–83.9) |
| | Yes | 1201 (39.5) | 1028 | 85.6 (80.8–90.4) |
| Employment type of household head | | | | |
| | Self-employed | 1452 (47.8) | 1162 | 80.0 (75.9–84.1) |
| | Paid worker | 1588 (52.2) | 1340 | 84.4 (80.2–88.6) |
| Marital status of household head | | | | |
| | Married | 2987 (98.3) | 2457 | 82.3 (79.3–85.3) |
| | Others[a] | 53 (1.7) | 45 | 84.9 (62.0–107.8) |
| Smoking of household head | | | | |
| | Yes | 286 (9.4) | 217 | 75.9 (67.1–84.7) |
| | No | 2754 (90.6) | 2285 | 83.0 (79.9–86.1) |
| Using narcotic drugs of household head | | | | |
| | Yes | 78 (2.6) | 63 | 80.8 (62.9–98.7) |
| | No | 2962 (97.4) | 2439 | 82.3 (79.3–85.3) |
| Age of mother/primary caregiver (years old) | | | | |
| | ≤17 | 29 (0.9) | 21 | 72.4 (46.0–98.8) |
| | 18–29 | 1843 (60.6) | 1585 | 86.0 (82.1–89.9) |
| | 30–44 | 969 (31.9) | 745 | 77.0 (72.2–81.8) |
| | 45≤ | 199 (6.6) | 151 | 75.8 (65.3–86.3) |
| Literacy of mother/primary caregiver | | | | |
| | Illiterate | 2308 (75.9) | 1983 | 80.5 (77.2–83.8) |
| | Literate | 732 (24.1) | 519 | 90.1 (83.6–96.6) |
| Listening to the radio by mother/primary caregiver | | | | |
| | No | 2199 (72.3) | 1817 | 82.6 (79.1–86.1) |
| | Yes | 841 (27.7) | 685 | 81.5 (76.0–87.0) |
| Watching TV by mother/primary caregiver | | | | |
| | No | 1841 (60.6) | 147 | 80.0 (76.3–83.7) |
| | Yes | 1199 (39.4) | 1029 | 85.8 (80.9–90.7) |

Penta3, the third dose of pentavalent vaccination.

[a]Others include single, divorced, widow, and widower.

(OR = 1.42, 95%CI 1.17–1.71), and had a mobile phone (OR = 1.40, 95%CI 1.11–1.77) had significantly more Penta3. In terms of variables of heads of households, having an education (OR = 1.47, 95%CI 1.20–1.79), being paid workers than self-employed (OR = 1.35, 95%CI 1.12–1.63), and non-smokers (OR = 1.54, 95%CI 1.61–2.07) were associated with Penta3. Children whose mothers/primary care givers were 17 years old or younger were negatively

associated with Penta3 compared to 18–29 years old (OR = 0.51, 95%CI 0.36–0.73). When mothers/primary caregivers were literate (OR = 2.51, 95%CI 2.05–3.05) or watched TV (OR = 1.51, 95%CI 1.24–1.84), the coverage of Penta3 among their children was significantly higher than the others.

Multivariable stepwise logistic regression analysis with forward-selection method (Model 1) using the 18 variables showed that factors associated with Penta3 among children aged 12–23 months old were 12–17 months old compared to 18–23 months old (AOR = 0.80, 95%CI 0.66–0.99); having no diarrhea in the last two weeks (AOR = 0.75, 95%CI 0.60–0.93) compared to having diarrhea; no bipedal edema compared to having edema (AOR = 0.43, 95%CI 0.22–0.85); taking vitamin A supplement (AOR = 1.72, 95%CI 1.34–2.22); 1–2 children under five years in a household compared to three or more (AOR = 2.06, 95% CI 1.67–2.54); distance from residence to the nearest health facility ≤2 hours on foot (AOR = 0.11, 95%CI 0.06–0.22); having a radio (AOR = 1.43, 95%CI 1.16–1.77); having a TV (AOR = 1.55, 95%CI 1.26–1.90); having educated heads of households (AOR = 1.65, 95%CI 1.32–2.05); non-smoking of heads of households (AOR = 1.50, 95%CI 1.09–2.06); and literacy of mothers/primary caregivers (AOR = 3.02, 95%CI 2.42–3.76) (Table 4). The factors associated with Penta3 in multivariable stepwise logistic regression analysis with backward-selection method (Model 2) were the same as those in Model 1 (Table 4).

## Discussion

This study showed that the coverage of Penta3 among 12–23-month-old children in Afghanistan was 82.3% in 2018, which was higher than that in the DHS 2015 (58%) [8, 12]. With recurrent outbreaks of pentavalent preventable diseases, the MoPH prioritized the immunization activities to strengthen the immunization systems and set the target of Penta3 vaccine coverage as 90% [5]. Health policies for promoting the equity of health services have been implemented and mobile immunization sessions have been expanded to areas with a low Penta3 coverage. However, the efforts of these policies were not so effective due to insecurity, absence of accurate data on target children, unavailability or low quality of monitoring and evaluation of the immunization activities, and vaccine hesitancy due to distrust in healthcare providers and the quality of health services [10, 12, 18, 22–24].

Better health status of children was associated with Penta3 coverage, namely no diarrhea in the last two weeks, no bipedal edema, and taking vitamin A supplement. These results were consistent with results of previous studies in other developing countries [19–21]. Bipedal edema is an indicator of severe acute malnutrition of children. Vitamin A is provided to 6–59-month-old children at health facilities at least twice a year by the National Immunization Campaigns and vitamin A deficiency weakens the immunity level of children against vaccine preventable diseases [19, 20]. These results suggest that parents who have appropriate knowledge of child health and sanitation are more likely to receive health services for children [19, 25–27]. This study also showed an association between distance (more than two hours on foot) from the nearest health facility and a lower vaccine coverage. Access to health service is also important for children's health, because poor access and distance from vaccination services were the most common reasons for not receiving Penta3 vaccines and hesitancy despite the availability of vaccines [10, 28].

The results of this study suggest that children whose parents were educated had a higher Penta3 coverage compared to those who had non-educated parents, because most household heads and caregivers were supposed to be fathers and mothers of the children. Previous studies reported that the education of parents strongly contributes to timely adherence to complete the schedule of Penta vaccine for children [28, 29]. Reading ability can promote understanding

**Table 4. Odds ratio and 95% confidence interval of Penta3 coverage among 12–23-month-old children (N = 3,040).**

| Variables | | OR (95% CI) | Model 1 | Model 2 |
|---|---|---|---|---|
| | | | AOR (95% CI) | AOR (95% CI) |
| Age of child (months old) | | | | |
| | 12–17 | 1 (Reference) | 1 (Reference) | 1 (Reference) |
| | 18–23 | 0.72 (0.60–0.87)** | 0.80 (0.66–0.99)* | 0.80 (0.66–0.99)* |
| Diarrhea in the last 2 weeks | | | | |
| | No | 1 (Reference) | 1 (Reference) | 1 (Reference) |
| | Yes | 0.71 (0.60–0.89)** | 0.75 (0.60–0.93)** | 0.74 (0.60–0.92)** |
| Bipedal edema | | | | |
| | Absent | 1 (Reference) | 1 (Reference) | 1 (Reference) |
| | Present | 0.45 (0.24–0.86)** | 0.43 (0.22–0.85)* | 0.44 (0.23–0.87)* |
| Vitamin A supplementation | | | | |
| | No | 1 (Reference) | 1 (Reference) | 1 (Reference) |
| | Yes | 1.80 (1.40–2.26)*** | 1.72 (1.34–2.22)*** | 1.69 (1.30–2.18)*** |
| Location of the household | | | | |
| | Rural | 1 (Reference) | - | 1 (Reference) |
| | Urban | 1.89 (1.44–2.49)*** | - | 1.29 (0.96–1.74) |
| Total eligible women | | | | |
| | 1 | 2.76 (2.07–3.68)*** | - | - |
| | 2≤ | 1 (Reference) | - | - |
| Total children under five years old | | | | |
| | 1–2 | 1.69 (1.40–2.05) | 2.06 (1.67–2.54)*** | 2.01 (1.63–2.45)*** |
| | 3≤ | 1 (Reference)*** | 1 (Reference) | 1 (Reference) |
| Type of toilet | | | | |
| | Sanitary | 1.41 (1.11–1.78)** | - | - |
| | Unsanitary | 1 (Reference) | - | - |
| Distance to the nearest health facility (on foot) | | | | |
| | ≤2 hr | 1 (Reference) | 1 (Reference) | 1 (Reference) |
| | 2 hr< | 0.07 (0.04–0.13)*** | 0.11 (0.06–0.22)*** | 0.12 (0.06–0.23)*** |
| Having a radio of household | | | | |
| | No | 1 (Reference) | 1 (Reference) | 1 (Reference) |
| | Yes | 1.30 (1.07–1.57)** | 1.43 (1.16–1.77)** | 1.42 (1.15–1.75)** |
| Having a television of household | | | | |
| | No | 1 (Reference) | 1 (Reference) | 1 (Reference) |
| | Yes | 1.42 (1.17–1.71)*** | 1.55 (1.26–1.90)*** | 1.48 (1.19–1.83)*** |
| Having a mobile phone of household | | | | |
| | No | 1 (Reference) | - | - |
| | Yes | 1.40 (1.11–1.77)** | - | - |
| Education of household head | | | | |
| | No | 1 (Reference) | 1 (Reference) | 1 (Reference) |
| | Yes | 1.47 (1.20–1.79)*** | 1.65 (1.32–2.05)*** | 1.58 (1.27–1.98)*** |
| Employment type of household head | | | | |
| | Self-employed | 1 (Reference) | - | 1 (Reference) |
| | Paid worker | 1.35 (1.12–1.63)** | - | 1.21 (0.99–1.49) |
| Smoking of household head | | | | |
| | Yes | 1 (Reference) | 1 (Reference) | 1 (Reference) |
| | No | 1.54 (1.61–2.07)** | 1.50 (1.09–2.06)* | 1.50 (1.10–2.06)* |
| Age of mother/primary caregiver (years old) | | | | |

*(Continued)*

**Table 4.** (Continued)

| Variables | | OR (95% CI) | Model 1 | Model 2 |
|---|---|---|---|---|
| | | | AOR (95% CI) | AOR (95% CI) |
| | ≤17 years | 0.51 (0.36–0.73)*** | 0.45 (0.19–1.05) | 0.46 (0.19–1.07) |
| | 18–29 years | 1 (Reference) | 1 (Reference) | 1 (Reference) |
| | 30–44 years | 0.85 (0.66–1.35) | 0.74 (0.31–1.75) | 0.74 (0.31–1.76) |
| | ≥45 years | 0.82 (0.38–1.72) | 0.76 (0.30–1.89) | 0.75 (0.30–1.86) |
| Literacy of mother/primary caregiver | | | | |
| | Illiterate | 1 (Reference) | 1 (Reference) | 1 (Reference) |
| | Literate | 2.51 (2.05–3.05)*** | 3.02 (2.42–3.76)*** | 3.00 (2.40–3.75)*** |
| Watching TV by mother/primary caregiver | | | | |
| | No | 1 (Reference) | - | - |
| | Yes | 1.51 (1.24–1.84)*** | - | - |

OR, odd ratio; AOR, adjusted odd ratio; CI, confidence interval.

Model 1: stepwise forward-selection (Hosmer-Lemeshow test: P = 0.451), Model 2: stepwise backward-selection (Hosmer-Lemeshow test: P = 0.101).

*P<0.05

**P<0.01

***P<0.001.

of the recommended immunization schedule, whereas illiteracy can decrease the Penta3 coverage [18, 25, 28, 30]. An association between receiving Penta3 and having a TV and a radio at home in this study suggests that household heads and mothers/caregivers might be able to receive health-related information even if they are illiterate. Announcements or dramas related to vaccination campaigns by TV or radio increases parents' awareness about benefits from vaccines [18, 23, 31, 32]. Furthermore, parents' attitude to their health may also affect Penta3 coverage because Penta3 coverage was significantly higher when household heads were non-smokers compared to when they were smokers in this study. When parents have proper knowledge of a healthy lifestyle, they understand the benefits of child immunization and their family members are healthy [18, 33].

The results of this study suggest that family characteristics such as the number of children in the household and age of mothers/primary caregivers may affect Penta3 coverage. It may be because a smaller family size can mitigate constraints in time allocation and resources [28, 34] and young mothers might have insufficient knowledge of child care because of early marriage, inadequate education, and lack of experience [35, 36]. In this study, children aged 12–17 months were significantly more likely to complete Penta3 compared to those aged 18–23 months. This may be because the Penta3 coverage has been increasing in Afghanistan. The improvement of the coverage from 2015 to 2018 was 24.3%, and it means 4.1% increase per six months. This is consistent with the result in this study that the difference of the coverage in the age group of 12–17 months old and the age group of 18–23 months old was 4.7%.

In the WHO Eastern Mediterranean region including 21 member countries, not only Afghanistan but also six other countries did not achieve the target of the DPT/Penta3 coverage among children under one year old in 2018; 87% in Afghanistan, 84% in Djibouti and Iraq, 80% in Yemen, 72% in Pakistan, 69% in Somalia, and 66% in Syrian Arab Republic [37]. These results suggest that living in conflict-affected areas may be a strong risk factor for low DPT/ Penta3 coverage in the region by causing cancellation of planned services and missed opportunities. Humanitarian crisis, such as wars and natural disasters, are one of major barriers for improving the vaccine coverage [38]. Identifying children in hard-to-reach areas and

expanding vaccine services to children who are older than the target age should be focused on more to improve the coverage.

Based on the results of this study, the following areas should be prioritized to increase Penta3 coverage in Afghanistan. First, health education, community awareness, and the formulation of ground-oriented interventions for social mobilization should be provided through different channels to increase vaccination knowledge and community demand for immunization. Second, primary education should be provided to both girls and boys, because they will be parents and the main decision-makers for child immunization. Third, people in hard-to-reach rural areas are vulnerable to vaccine-preventable diseases; therefore, regular mobile outreach programs for these communities should be established through routine immunization outreach service delivery plans. Fourth, the location and equity distribution of health facilities should be reassessed, and new health facilities should be established. This is pertinent because a distance of more than two hours on foot from health facilities is one of the main barriers for Penta3 vaccine completion.

This study has some limitations. First, variables included in this study did not include some important information such as the wealth status of the children's family or characteristics of parents but not household heads or primary caregivers [5, 12, 21, 39, 40], because this study used the data of the AHS 2018 for secondary analysis. Second, the immunization history was not based on the immunization cards of all children but based on the self-reports of mothers/primary caregivers and household heads of 11.2% of all the children. Therefore, recall bias might occur when parents answered without immunization cards and the Penta3 coverage might be over- or under-estimated. Third, some areas were not included in this study due to security reasons and Penta3 coverage might be lower than the coverage estimated in this study. However, all 34 provinces were covered and 3,040 children were included in this study. The results in this study may be useful to make policies for improving the vaccination coverage in Afghanistan.

## Conclusion

In conclusion, the coverage of Penta3 among children aged 12–23 months in Afghanistan was 82.3% in 2018, which showed improvement but did not reach the target coverage. Factors associated with Penta3 coverage were 12–17 months old compared to 18–23 months old, having no diarrhea in the last two weeks, no bipedal edema, taking vitamin A supplements, 1–2 children under five years in a household compared to three or more, distance from residence to the nearest health facility ≤2 hours on foot, having a radio, having a TV, education and non-smoking of heads of households, and literacy of mothers/primary care givers. Primary education should be provided to both boys and girls throughout the country and information about vaccination schedules and health should be provided by TV and radio. To improve the access to health service, promoting mobile outreach programs and establishing new health facilities are also recommended.

## Acknowledgments

We are grateful to the Health Information System General Department and the EPI Department of the MoPH for their support and assistance in providing the dataset of the AHS 2018. We also appreciate Dr. Asmatullah Arab, Dr. Werishmen Sabawoon, and Mr. Sadiq Mossadiq for their insights and technical support.

## Author Contributions

**Conceptualization:** Muhammad Kamel Frozanfar, Nobuyuki Hamajima, Eiko Yamamoto.

**Data curation:** Muhammad Kamel Frozanfar, Hedayatullah Stanekzai.

**Formal analysis:** Muhammad Kamel Frozanfar, Nobuyuki Hamajima, Eiko Yamamoto.

**Methodology:** Muhammad Kamel Frozanfar, Nobuyuki Hamajima, Eiko Yamamoto.

**Supervision:** Nobuyuki Hamajima, Eiko Yamamoto.

**Writing – original draft:** Muhammad Kamel Frozanfar.

**Writing – review & editing:** Muhammad Kamel Frozanfar, Nobuyuki Hamajima, Said Hafizullah Fayaz, Abdullah Darman Rahimzad, Souphalak Inthaphatha, Kimihiro Nishino, Eiko Yamamoto.

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
