## [Decision Letter · Decision Letter 0]

26 Dec 2022

PONE-D-22-30048Factors associated with pentavalent vaccine coverage among 12–23-month-old children in Afghanistan: A cross-sectional studyPLOS ONE

Dear Dr. Yamamoto,

Thank you for submitting your manuscript to PLOS ONE. After careful consideration, we feel that it has merit but does not fully meet PLOS ONE’s publication criteria as it currently stands. Therefore, we invite you to submit a revised version of the manuscript that addresses the points raised during the review process.

We look forward to receiving your revised manuscript.

Kind regards,

Orvalho Augusto, MD, MPH

Academic Editor

PLOS ONE

Journal Requirements:

2. You indicated that you had ethical approval for your study. In your Methods section, please ensure you have also stated whether you obtained consent from parents or guardians of the minors included in the study or whether the research ethics committee or IRB specifically waived the need for their consent. Furthermore, you have specified that verbal consent was obtained. Please provide additional details regarding how this consent was documented and witnessed, and state whether this was approved by the IRB.

Reviewers' comments:

Reviewer's Responses to Questions

**Comments to the Author**

1. Is the manuscript technically sound, and do the data support the conclusions?

Reviewer #1: Partly

2. Has the statistical analysis been performed appropriately and rigorously? 

Reviewer #1: Yes

3. Have the authors made all data underlying the findings in their manuscript fully available?

Reviewer #1: No

4. Is the manuscript presented in an intelligible fashion and written in standard English?

Reviewer #1: Yes

5. Review Comments to the Author

Reviewer #1: This study “Factors associated with pentavalent vaccine coverage among 12–23-month-old children in Afghanistan: A cross-sectional study” describes the factors associated with administration of the third dose of pentavalent vaccine (Penta3) among children aged 12–23 months in Afghanistan, based on 2018 Afghanistan Health Survey data. The number of eligible children was 3,040 corresponding 19,684 households who had a child aged 12-23 months, the information was obtained from child mothers or a legal tutor. The coverage of Penta3 among 12–23-month-old children was 82.3%. Important determinants were identified associated with Penta3 administration such as age, having no diarrhea in the last two weeks, no bipedal edema, taking vitamin A supplement, household members, distance from residence to the nearest health facility, having a radio, having a TV, having education of heads of households, non-smoking of heads of households, age of mothers/primary care givers, and literacy of mothers/primary caregivers.

The findings reported in this manuscript fit the aims and scope of the journal; however, it does not present novel findings as the previous analysis was performed using the Afghanistan Health Survey 2012/2015 and the factors and findings do not differ too much. The authors should show the relevance of the continuous analysis of the surveys in order to monitor the determinants. I suggest major revision and clarifications for specific points.

Introduction and methods

What is the difference between administration and coverage? Please provide the definition of vaccine coverage in your introduction.

Just to reinforce, it’s a suggestion: Please provide information about the EPI centers, their establishment and operations, if vaccine administration is provided free of charge by the state.

UNICEF/WHO target of immunization is not mentioned in the manuscript, add it please.

L65-66: suggestion change (ages 1-4) by (under five years).

L67: please use updated information if available, 2015 to today many things happened.

L78-80: Please provide the national vaccination coverage estimate. If possible, show how it improved from 2015 to 2018 and wish regions of Afghanistan present the lowest coverage.

L80-83: suggestion: The factors associated with non- and under-vaccination of children in Afghanistan were reported to be socio-demographic and childhood factors, lack of urbanization, access to health facilities, and household factors including maternal and paternal education.

L83-85: The justification can be improved, data regarding vaccination coverage using community data are available in Afghanistan, it would be important to provide a strong justification and show which gap your study will fill.

L84: you have Penta3 administration, however, in L86 coverage, please standardize.

Materials and methods

For better visualization please provide a map that locates it on the continent, as well as the division by provinces.

The time of data collection is not mentioned by the authors the survey took a years or a couple of months? How did they collect it? by visiting individual home? What was the response rate of the participating mothers? method section lacks this information.

L91-92: Change by Cross-sectional study, conducted by the MoPH……………

L98-99: please make it clear

L116-L118: Suggestion: The characteristics of the 12–23-month-old children, their households, characteristics of their household-heads and mothers/caregivers were taken from the AHS database.

L124-125: In Afghanistan, the Vitamin A administration is made only in National campaigns and not as a routine immunization in Health Center? And this information usually goes to child health cards, why this information was only obtained through interviews. Please check L296

L133: please check if is “number of total eligible women” or “number of total eligible children”.

L153-162 and 163-170 check if the information is not overlapped, what is the difference between these two paragraphs and why they are important?

L173 there is no information about the descriptive analysis, please add.

L187: something is missing after the comma

190: please remove the word “old”

191: Please check the percentage of children without vaccination card, because in line 191 you have 11.2%, while in the limitation section 12.0%, please revise.

Results

i. General observation: how did you conduct the descriptive analysis on the tables? I was expected to see each one of the values divided by the sample size, not by its own “n”. For example, in table 1, I was expected to see diarrhea variable 68.4% for “no” and “31.6%” for yes, which reflects the estimates that you have in the country survey. This is just an observation the authors can think and discuss about it.

191-196: I think it deserves a subtitle maybe “Vaccination coverage” and concentrate all the information regarding the vaccine coverage.

Discussion

General

Make a comparative assessment of vaccination coverage in Afghanistan and other countries in the region.

6. PLOS authors have the option to publish the peer review history of their article (what does this mean?). If published, this will include your full peer review and any attached files.

Reviewer #1: No

---

## [Author Response · Author response to Decision Letter 0]

1 Feb 2023

Response to editor’s and reviewers’ comments 

We would like to thank the editor and the reviewers for reviewing our manuscript. We have revised the manuscript according to your comments, which were very helpful. The revised manuscript has been proofread by a native English speaker. The revisions have been completed and the responses are as follows.

Journal Requirements: 

We have carefully revised our manuscript to meet PLOS ONE’s style requirements including file names.

2. You indicated that you had ethical approval for your study. In your Methods section, please ensure you have also stated whether you obtained consent from parents or guardians of the minors included in the study or whether the research ethics committee or IRB specifically waived the need for their consent. Furthermore, you have specified that verbal consent was obtained. Please provide additional details regarding how this consent was documented and witnessed, and state whether this was approved by the IRB.

Lines 206-212: In the AHS 2018, a verbal informed consent form was used during the data collection. The interviewers read all contents of the form to each participant. When they obtained verbal consent from each participant, they wrote their name and the date of interview, checked the box of “consented”, and signed on the form. When the interviewers collected information about children, they obtained verbal consent from the children’s mother/primary caregiver. We have revised the paragraph of “Ethical issues” as follows, “Verbal consent was obtained from each participant, namely a household head and a mother/primary caregiver of each child, after interviewers read all contents of a verbal consent form to the participant. For data collection of a child, verbal consent was obtained from the mother/primary caregiver of the child. When verbal consent was obtained, the interviewers checked the box of “consented”, wrote their name and the date of interview, and signed on the consent form.”

The data for AHS 2018 is a property of the Ministry of Public Health (MoPH) of Afghanistan. MoPH of Afghanistan has the ownership and authority to manage the AHS 2018 data. Only authorized and permitted researchers by MoPH are allowed to access and use the data for research and analytical purposes. Data are available from the Ministry of Public Health of Afghanistan (General Directorate of Monitoring & Evaluation and Health Information System; Tel +93-20-231-3688 | Emails ehis.moph@gmail.com, mehis.moph@gmail.com) for researchers who meet the criteria for access to confidential data.

Reviewer #1:

Introduction

1. What is the difference between administration and coverage? Please provide the definition of vaccine coverage in your introduction.

We used the term “administration” as the process of applying vaccine for the targeted children and the term “coverage” as the outcome of applying vaccine for the target children. We think that “coverage” is more appropriate rather than “administration” in our manuscript and we have revised “administration” to “coverage” in the manuscript.

2. Just to reinforce, it’s a suggestion: Please provide information about the EPI centers, their establishment and operations, if vaccine administration is provided free of charge by the state.

Line 80-83: Immunization is one of the key components of Basic Package of Health Services in Afghanistan and provided at EPI centers free of charge. EPI centers have at least one trained vaccinator to apply the routine immunization vaccines to under-five children on a daily basis. The number of EPI centers increased from 1,575 in 2015 to 2,926 in 2018 and EPI centers cover all areas in the country. We have added as follows, “Since the EPI started in Afghanistan, the vaccine delivery system has been improving. EPI centers, which have at least one trained vaccinator, provide the routine immunizations free of charge and the number of EPI centers increased from 870 in 2004 to more than 1,767 in 2016 [13].”

3. UNICEF/WHO target of immunization is not mentioned in the manuscript, add it please.

Lines 76-78: The WHO Eastern Mediterranean Region (EMRO) set the 90% vaccination target for member countries. We have added a sentence as follows, “The WHO Eastern Mediterranean Region set the target of childhood vaccine coverage for all antigens as 90% for countries in the region, including Afghanistan [11].”

4. L65-66: suggestion change (ages 1-4) by (under five years).

Line 65-66: We have revised “child mortality rate (CMR) (ages 1-4)” to “mortality rate of children under five years old (under-five mortality).”

5. L67: please use updated information if available, 2015 to today many things happened.

Line 66-68: The World Bank estimates the mortality rate of children under five years old in Afghanistan and the latest estimation reported 58 per 1,000 live births in 2020. We have revised the sentence as follows, “the under-five mortality in Afghanistan declined from 191 per 1,000 live births in 2006 to 58 per 1,000 live births in 2020 [8,9].”

6. L78-80: Please provide the national vaccination coverage estimate. If possible, show how it improved from 2015 to 2018 and wish regions of Afghanistan present the lowest coverage.

Line 80: “Immunization services” means “vaccine delivery system” and we have revised as follows, “the vaccine delivery system has been improving.” 

Lines 83-85: According to the results of the nation-wide surveillance, the routine immunization coverage increased from 30.1% in 2003 to 59.7% in 2013. However, the coverage was 58.6% in 2015, which decreased to 50.2% in 2018. The coverage was especially low in southern and southeastern provinces (Helmand, Zabul, Urozgan, and Paktika Provinces) (3.1%-28.1%). We have added a sentence as follows, “The routine immunization coverage improved from 30.1% in 2003 to 59.7% in 2013 [10], but decreased from 58.8% in 2015 and 50.2% in 2018 [8, 14]. The coverage in 2018 was low (3.1-28.1%) in southern and southeastern provinces.”

7. L80-83: suggestion: The factors associated with non- and under-vaccination of children in Afghanistan were reported to be socio-demographic and childhood factors, lack of urbanization, access to health facilities, and household factors including maternal and paternal education.

Line 92-95: According to the reviewer’s comment, we have revised as follows, “the factors associated with non- and under-vaccination of children in Afghanistan were socio-demographic and childhood factors, lack of urbanization, access to health facilities, and household factors including maternal and paternal education.”

8. L83-85: The justification can be improved, data regarding vaccination coverage using community data are available in Afghanistan, it would be important to provide a strong justification and show which gap your study will fill.

Line 80-98: The vaccine service system has been developing, but Afghanistan has still outbreaks of infectious diseases which can be prevented by the 3 doses of pentavalent vaccine (Penta3). Vaccine service by mobile health teams was introduced in 2003 but the coverage of Penta3 was not higher in the areas with mobile health team services compared to the areas without the services. Therefore, factors influencing lower coverage of Penta 3 should be understood but no studies have been published focusing on factors associated with Penta3 coverage among children in Afghanistan. We have revised the part of the paragraph as follows, “Since the EPI started in Afghanistan, the vaccine delivery system has been improving. EPI centers, which have at least one trained vaccinator, provide the routine immunizations free of charge and the number of EPI centers increased from 870 in 2004 to more than 1,767 in 2016 [13]. The routine immunization coverage improved from 30.1% in 2003 to 59.7% in 2013 [10], but decreased from 58.8% in 2015 and 50.2% in 2018 [8, 14]. The coverage in 2018 was low (3.1-28.1%) in southern and southeastern provinces. Therefore, Afghanistan still has strong outbreaks of vaccine-preventable diseases, especially diseases that can be prevented by Penta. In 2003, providing maternal and child health services by mobile health teams, which consist of a midwife, a vaccinator, and a nurse, started in remote and conflict-affected areas. In 2016 and 2017, the coverage of Penta3 in the areas with mobile health team services was 26.6%, which was lower than that in the areas without the services (30.5%) [15]. To improve the coverage of Penta3, the factors affecting no or incomplete vaccination of Penta3 should be understood. Previous studies reported that the factors associated with non- and under-vaccination of children in Afghanistan were socio-demographic and childhood factors, lack of urbanization, access to health facilities, and household factors including maternal and paternal education [2,5,12]. However, to our knowledge, there have been no studies on the factors associated with Penta3 coverage in Afghanistan. Therefore, this study aimed to identify factors associated with Penta3 coverage among children aged 12-23 months in the country, using the data of the Afghanistan Health Survey (AHS) 2018.”

9. L84: you have Penta3 administration, however, in L86 coverage, please standardize.

Line 96: As we have stated to a previous comment above, we have revised “administration” to “coverage” throughout the manuscript.

Materials and Methods:

10. For better visualization please provide a map that locates it on the continent, as well as the division by provinces.

Line 104: Thank you very much for your comment. We use “Afghanistan Health Survey 2018” as a reference (No. 14), which explains the sampling methods and includes maps of provinces of Afghanistan. Therefore, everyone can see the maps by searching “Afghanistan Health Survey 2018” without a figure of a country map in the manuscript. For this reason, we do not feel that it is necessary to include a figure of a map.

11. The time of data collection is not mentioned by the authors the survey took a years or a couple of months? How did they collect it? by visiting individual home? What was the response rate of the participating mothers? method section lacks this information.

Lines 109-110: Data collection took place between March 2018 and August 2018. Field teams consisted of male and female interviewers who visited each selected household and interviewed all men who have ever been and women in the household after obtaining verbal consent. The report of AHS showed the response rate of households and women and the complete rate of children’s data (anthropometric measurement and health data). Furthermore, children included in the AHS 2018 were under five years old but our study included children aged 12–23 months old. Therefore, we could not find the response rate of mothers/primary care givers of children 12-23 months old. We have revised the method section as follows, “Data collection was conducted from March to August in 2018 [14]. Field teams consisted of male and female interviewers, recruited from each of the 34 provinces of Afghanistan. All staff received extensive training prior to going into the field. The AHS instruments consisted of six questionnaires: (1) general information and socio-demography, (2) household history, (3) anthropometry of children, (4) mother’s general information, (5) women’s pregnancy history, and (6) child immunization status. The field teams visited each selected household and interviewed all men who had ever been married and women aged 15-49 of the household using the questionnaires after obtaining verbal consent. The response rate of households and women was 94.4% and 93.9%, respectively [14]. Regarding children’s data collection, the anthropometric completion rate and child health completion rate was 93.8% and 94.3%, respectively. Data collection in the field took place by adequate interview techniques maintaining all ethical protocols. When the data were analyzed, sampling weights were applied in order to adjust the results for differences that might be caused by the selection steps.”

12. L91-92: Change by Cross-sectional study, conducted by the MoPH……………

Line 102: The manuscript has been proofread by a native English speaker. According to the English editor’s advice, it would be better to keep the original sentence. 

13. L98-99: please make it clear

Line 116-120: We understand that the reviewer asked about “for security reasons and the safety of data collectors.” The security issues were recorded active fights (civil wars) in the selected clusters; therefore, female team members were not allowed to enter the area. The survey team was not allowed to use electronic devices. Furthermore, access to the area was difficult because of the geographical terrain. We have revised the sentence as follows, “However, less than two-thirds of the selected clusters were surveyed in Zabul, Nuristan, and Helmand Provinces due to civil wars and hard-to-reach areas [14].”

14. L116-L118: Suggestion: The characteristics of the 12–23-month-old children, their households, characteristics of their household-heads and mothers/caregivers were taken from the AHS database.

Lines 138-140: The manuscript has been proofread by a native English speaker. According to the English editor’s advice, it would be better to keep the original sentence.

15. L124-125: In Afghanistan, the Vitamin A administration is made only in National campaigns and not as a routine immunization in Health Center? And this information usually goes to child health cards, why this information was only obtained through interviews. Please check L296

Lines 149-150: In Afghanistan, vitamin A is provided to children by the National Immunization Campaigns but not the routine immunization. The child vaccination card for routine immunization in Afghanistan does not record the administration of vitamin A. Therefore, data collectors asked mothers/primary caregivers if their child took vitamin A or not. We have added as follows, “In Afghanistan, vaccine record cards do not record vitamin A supplementation.”

16. L133: please check if is “number of total eligible women” or “number of total eligible children”.

Line 156: We have checked and “number of total eligible women” is correct.

17. L153-162 and 163-170 check if the information is not overlapped, what is the difference between these two paragraphs and why they are important?

Lines 176-185 and 186-193: The first paragraph (lines 176-185) explains about variables for the data of the heads of household and the second paragraph (lines 186-193) explains about variables of the data mothers/primary caregivers. Age and the educational history were included in both data but the methods of categorization were different. Variables other than age and the educational level were different between household heads and mothers/primary caregivers. Therefore, these two paragraphs are needed and important to explain about variables and categories.

18. L173 there is no information about the descriptive analysis, please add.

Lines 196-197: We have added the sentence as follows, “Descriptive analysis was used to describe the characteristics of children, their households, and their household heads and mothers/caregivers.”

Results:

19. L187: something is missing after the comma

Line 215: We have checked the sentence carefully and there is nothing missing after the comma.

20. 190: please remove the word “old”

Lines 217-218: “1,742 children (57.3%) were 12–17 months old” and “the average age was 16.5 months old” are grammatically correct.

21. 191: Please check the percentage of children without vaccination card, because in line 191 you have 11.2%, while in the limitation section 12.0%, please revise.

Line 387: We are sorry that the percentage in the limitation section was incorrect. We have revised it from 12.0% to 11.2%.

22. General observation: how did you conduct the descriptive analysis on the tables? I was expected to see each one of the values divided by the sample size, not by its own “n”. For example, in table 1, I was expected to see diarrhea variable 68.4% for “no” and “31.6%” for yes, which reflects the estimates that you have in the country survey. This is just an observation the authors can think and discuss about it.

Tables 1-3: The percentage of “Penta3” shows the proportion of children who had Penta3 among total children in each category. We have added the number and percentage of children who did not have Penta3 (who had 0-2 doses of pentavalent vaccine).

23. 191-196: I think it deserves a subtitle maybe “Vaccination coverage” and concentrate all the information regarding the vaccine coverage.

This is a part of the result section and subtitles are not used in the result section. The percentage of columns of “Penta3” means the coverage of Penta3 among each category in Tables 1-3. The comparison of the coverage among categories of each variable is explained from page 9 to page 12.

Discussion

24. Make a comparative assessment of vaccination coverage in Afghanistan and other countries in the region.

Lines 360-369: In our study, the Penta3 coverage among children 12-23 months old. It is difficult to find the same coverage in other countries. Therefore, we compared the coverage of DPT/Penta3 coverage among 21 member countries in the WHO Eastern Mediterranean region using the data by the WHO. We have added a paragraph in the discussion section as follows, “In the WHO Eastern Mediterranean region including 21 member countries, not only Afghanistan but also six other countries did not achieve the target of the DPT/Penta3 coverage among children under one year old in 2018; 87% in Afghanistan, 84% in Djibouti and Iraq, 80% in Yemen, 72% in Pakistan, 69% in Somalia, and 66% in Syrian Arab Republic [37]. These results suggest that living in conflict-affected areas may be a strong risk factor for low DPT/Penta3 coverage in the region by causing cancellation of planned services and missed opportunities. Humanitarian crisis, such as wars and natural disasters, are one of major barriers for improving the vaccine coverage [38]. Identifying children in hard-to-reach areas and expanding vaccine services to children who are older than the target age should be focused on more to improve the coverage.”

---

## [Decision Letter · Decision Letter 1]

16 May 2023

PONE-D-22-30048R1Factors associated with pentavalent vaccine coverage among 12–23-month-old children in Afghanistan: A cross-sectional studyPLOS ONE

Dear Dr. Yamamoto,

Thank you for submitting your manuscript to PLOS ONE. After careful consideration, we feel that it has merit but does not fully meet PLOS ONE’s publication criteria as it currently stands. Therefore, we invite you to submit a revised version of the manuscript that addresses the points raised during the review process.

We look forward to receiving your revised manuscript.

Kind regards,

Orvalho Augusto, MD, MPH

Academic Editor

PLOS ONE

Journal Requirements:

Additional Editor Comments:

This is an important manuscript documenting 1) the level of the third dose of the pentavalent vaccine in Afghanistan, and 2) potential factors. Thank you for responding to the reviewers' questions. There still remain some shortcomings to resolve:

1. Please make subdivisions on the abstract as one reviewer points out.

2. Line 68 there is one citation pointing to the World Bank as the source of child mortality estimates. This is incorrect. Please a UN IGME reference for this.

3. About the methods:

- About the AHS 2008 sampling. With the current description, it seems that there is province stratification. How about urban/rural?

- Line 113/114: the last sentence is a repetition of a sentence in 110/112. Please, revise.

- There were 3 provinces that did not reach the desired sample size. What was done for these provinces at the analysis level?

- What languages were used to talk to the mothers? Were the data collectors trained for this?

- In a household with more than 1 child eligible to enter the sample what was done? This is an important detail to describe.

- Did the analysis use survey weights or not? This seems to be ignored in the whole analysis.

- Line 180: The categorization of age obeys quantiles of age. What this means quartiles, quintiles? The below age 29 may include families led by someone under-18 or under-21. This may matter at least for description.

- About the "ethical issues". This report is a result of a secondary data analysis of AHS 2018, correct? If yes, clarify that the ethical approval you got was to access the data not to conduct the data collection.

4. Results:

- Please do not use only p-values on table 1 through table 3 (it would be fine to remove these p-values). In fact, many of these p-values are repeated in the unadjusted analysis of table 4. For tables 1 to 3 do: 1) make the percentages to be by column; 2) add a column for coverage and its 95% confidence interval.

- Table 4 - there is nowhere in the manuscript explaining how the factors were chosen. Is this based on the literature? Please state so. Why not adjust by province?

Reviewers' comments:

Reviewer's Responses to Questions

**Comments to the Author**

1. If the authors have adequately addressed your comments raised in a previous round of review and you feel that this manuscript is now acceptable for publication, you may indicate that here to bypass the “Comments to the Author” section, enter your conflict of interest statement in the “Confidential to Editor” section, and submit your "Accept" recommendation.

Reviewer #2: (No Response)

Reviewer #3: (No Response)

2. Is the manuscript technically sound, and do the data support the conclusions?

Reviewer #2: Yes

Reviewer #3: Yes

3. Has the statistical analysis been performed appropriately and rigorously? 

Reviewer #2: Yes

Reviewer #3: Yes

4. Have the authors made all data underlying the findings in their manuscript fully available?

Reviewer #2: Yes

Reviewer #3: No

5. Is the manuscript presented in an intelligible fashion and written in standard English?

Reviewer #2: Yes

Reviewer #3: Yes

6. Review Comments to the Author

Reviewer #2: Thanks for allowing me to review this manuscript. Over all the manuscript is well written but I have some concerns and suggestions as stated below.

1. Title: The title is good but why you prefer only coverage of PCV3 vaccine coverage? Why you ignore PCV1, PCV2….???

2. Your source of data is not clear; is it primary data from face to face interview or secondary data from previous records??? Be consistent and revise it throughout the document. If you say it was secondary data how do you get consent from participants?

3. The variables like diarrhea, bipedal edema, vitamin A supplement, smoking of heads of households,… are irrelevant and clinically insignificant variables for vaccine coverage so why you include this factors on multivariable regression? Because inclusion of such irrelevant variables in the analysis might confound the effect of other variables.

4. Variables of the study should better to be summarized under one or two paragraphs and operational definitions should be stated in separate section.

5. How do you check multicollinarity? Better to incorporate it in the manuscript.

6. Almost all tables are mere repetitions of text descriptions, better to avoid redundancy by choosing either of the two.

7. Variables such as; diarrhea, bipedal edema, vitamin A supplement and smoking of heads of households are not strongly justified for its association with vaccine coverage in the discussion section. revise it again

Reviewer #3: Thank you for the invitation to review this paper. This paper worth publishing. However, it has some shortcoming from abstract to reference.

The abstract section lacks sub heading. The abstract section is lengthy.

The introduction section is also too long. It should address the tried effort to decrease the burden of the problem your country, current intervention, and any future directions. Furthermore, you should give a strong gap. Your gap is not convincing.

In the method section, you merged method sub-sections. You need to separate and present in clear way. Data collection method, sampling procedure, Tool.

There are a lot of repetitive ideas. E.g. the study period is presented in page 3 line number 103, 115....

Is your tool valid and reliable? Is it developed by the authors or adopted?

Check the whole document for typo errors. Be consistent with use of grammars. Statistical analyses vs statistical analysis?

In the result section: you need to avoid repetition of data already stated in the table. Make it short and precise.

Strong recommendation: The result section should be written in short and precise way.

Have you checked the model assumptions? if so report the findings.

In the discussion section: please check the typing errors. You need to discuss the implication of the finding.

The declaration section seems incomplete.

7. PLOS authors have the option to publish the peer review history of their article (what does this mean?). If published, this will include your full peer review and any attached files.

Reviewer #2: No

Reviewer #3: No

---

## [Author Response · Author response to Decision Letter 1]

22 May 2023

Response to editor’s and reviewers’ comments 

We would like to thank the editor and the reviewers for reviewing our manuscript. We have revised the manuscript according to your comments, which were very helpful. The revised manuscript has been proofread by a native English speaker. The revisions have been completed and the responses are as follows.

Journal Requirements: 

We are very sorry that we did not explain about changing references or adding references when we submitted revised manuscript. It is because we used EndNote (a software for references). When EndNote is used for references, revisions of references and reference numbers cannot be found even if the function of “track change” is used in WORD file. Therefore, we have added underlines for the revised reference numbers in the manuscript and for the revised reference (No. 9) in the reference section. 

Additional Editor Comments:

2. Please make subdivisions on the abstract as one reviewer points out.

We have added subheadings in the abstract.

3. Line 68 there is one citation pointing to the World Bank as the source of child mortality estimates. This is incorrect. Please a UN IGME reference for this.

Line 68 and reference No. 9: We have revised the reference (No. 9) for under-five mortality rate in Afghanistan from the World Bank to United Nations Inter-agency Group for Child Mortality Estimation.

Methods:

4. About the AHS 2018 sampling. With the current description, it seems that there is province stratification. How about urban/rural?

Lines 106, 109-110: We have added the explanation of stratification considering urban and rural areas and the number of households in urban and rural area as follows, “Stratification was achieved by separating each province into urban and rural areas. “ (line 106) and “Therefore, 23,460 households (4,738 households in urban areas and 18,722 households in rural areas) were selected as the survey samples in all 34 provinces.” (lines 109-110).

5. Line 113/114: the last sentence is a repetition of a sentence in 110/112. Please, revise.

We have found that two sentences were repeated in the paragraph and we have erased them. 

6. There were 3 provinces that did not reach the desired sample size. What was done for these provinces at the analysis level?

We applied sampling weights when we performed analyses, because sampling weights were calculated according to two types of probability of selection, including selection by design and by factors other than design (affected by non-response of a respondent). Taking into account response rates in the weights, makes it also inevitable to introduce individual weights, in addition to household weights. The fact that the questionnaires were designed to interview different types of respondents for household, eligible women and children under 5, anthropometric, and verbal autopsy sections, requires separate response rates for these types of respondents. In AHS, these response rates are calculated at the stratum level. The combination of these two factors, design and non-response, produces household and different individual weights.

7. What languages were used to talk to the mothers? Were the data collectors trained for this? 

Lines116, 186-188: The questionnaire forms were translated to Dari and Pashto. When literacy of mothers/primary caregivers was estimated, the data collectors observed their reading and writing ability of any of the common languages of the country (Pashto, Dari, or Uzbek language). Extensive training was provided to trainers and field staff prior to going into the field. The detail methods of trainings of data collectors are explained in the report of AHS 2018, which is included as a reference in our manuscript (No. 14). 

8. In a household with more than 1 child eligible to enter the sample what was done? This is an important detail to describe. 

Lines 131-133: All households included in the AHS 2018 had only one eligible child (12-23-month old). We explained as follows, “Of the 19,684 households included in the AHS 2018, 3,040 households had a child aged 12–23 months and there was no household that had two or more children aged 12–23 months.”

9. Did the analysis use survey weights or not? This seems to be ignored in the whole analysis.

Yes, we used sampling weights in the analyses. 

10. Line 180: The categorization of age obeys quantiles of age. What this means quartiles, quintiles? The below age 29 may include families led by someone under-18 or under-21. This may matter at least for description.

We divided age of household heads into the four groups and the interval was 15 years. Each category suggests characteristics of people. “≤29 years” means the young generation, “30–44 years” means the generation of reproductive, active, and having experiences, “45–59 years” means “senior but still active”, and “≥60” means “after retirement and grandparents”. It is not so rare in Afghanistan that household heads are under18 years or 21 years because of conflicts or terrorist attacks. In our study, household heads who were under 18 years and 21 years were 8 (0.3%) and 40 (1.3%), respectively. In lines 248-249, we have added a sentence as follows, “There were 8 (0.3%) and 40 (1.3%) household heads who were under 18 and 21 years, respectively.”

11. About the "ethical issues". This report is a result of a secondary data analysis of AHS 2018, correct? If yes, clarify that the ethical approval you got was to access the data not to conduct the data collection.

Lines 138-139, 207-209: This study conducted a secondary analysis of the data of AHS 2018 and we did not do data collection. Our study was approved by IRB of the MoPH in Afghanistan. The first author used to work for the MoPH in Afghanistan. For his PhD study, he requested the AHS dataset to General Directorate of Monitoring & Evaluation and Health Information System, MoPH and it was accepted. In lines 138-139, we have added the following sentence, “The dataset was provided by the MoPH for this study.”

Results:

12. Please do not use only p-values on table 1 through table 3 (it would be fine to remove these p-values). In fact, many of these p-values are repeated in the unadjusted analysis of table 4. For tables 1 to 3 do: 1) make the percentages to be by column; 2) add a column for coverage and its 95% confidence interval.

Tables 1-3: This style of tables and a combination of chi-square test/Fisher’s exact test and logistic regression analysis are commonly used in many papers that were published not only from PLoSOne but also other journals. The coverage of Penta3 in each category is easy to understand in Tables 1-3, because it is written as the percentage of children in the column of Penta 3 “Yes.”

13. Table 4 - there is nowhere in the manuscript explaining how the factors were chosen. Is this based on the literature? Please state so. Why not adjust by province?

Lines 192-193: The variables were chosen according to previous studies on vaccination coverage. We have added the following sentence, “Variables for this study were chosen according to previous studies on the vaccine coverage [2, 5, 12, 18-21].” 

There are 34 provinces in Afghanistan. We would like to perform analyses to understand the vaccine coverage and associated factors in the national level but the provincial level. Therefore, province was not included in variables in our study.

Reviewer #2:

14. Title: The title is good but why you prefer only coverage of PCV3 vaccine coverage? Why you ignore PCV1, PCV2….???

It is because three doses of pentavalent vaccine are recommended and scheduled in the EPI schedule in Afghanistan.

15. Your source of data is not clear; is it primary data from face to face interview or secondary data from previous records??? Be consistent and revise it throughout the document. If you say it was secondary data how do you get consent from participants?

Line 102: We have moved the sentence “This study is a secondary analysis of the AHS 2018 data” to the beginning of the method section, to emphasize that this study conducted a secondary analysis. 

Lines 207-209: In the AHS, informed consent was obtained from each participant in the AHS. In this study, informed consent was waived due to a secondary analysis of anonymous data of the AHS and the study protocol of this study was approved by IRB. We have revised the sentence as follows, “In this study, written informed consent was waived due to a secondary analysis of the anonymous data of the AHS 2018 and the study protocol was approved by the Institutional Review Board of the MoPH in Afghanistan (Approval number: E.1019.0087).”

16. The variables like diarrhea, bipedal edema, vitamin A supplement, smoking of heads of households,… are irrelevant and clinically insignificant variables for vaccine coverage so why you include this factors on multivariable regression? Because inclusion of such irrelevant variables in the analysis might confound the effect of other variables.

In a setting like Afghanistan, children’s nutrition status and immunity level, and parents’ behavior and habits are highly connected with immunization and health status of under five children. These are reported in various studies and reports in Afghanistan and other developing countries. Furthermore, in AHS 2018, the immunization coverage of the children was evaluated and was associated with the vitamin A supplementation, diarrhea history and nutrition status of the children. In various reports and studies, parents’ habits such as smoking were associated with education level, child health statues, immunization, and health practices in the household. Therefore, we believe that it is necessary to include household factors (for example; smoking cigarettes by the head of the household), because the results increase the awareness of improving desirable behavior and habits among the parents and give an alert to the Ministry of Health of the country and legislation authorities. It is easy and cheap to purchase (2 or more boxes of cigarettes cost 1 USD) and smoke cigarettes without considering any public health and community norms. Considering these facts and results, we included diarrhea, bipedal edema, vitamin A supplement, smoking of heads of households in this study. 

17. Variables of the study should better to be summarized under one or two paragraphs and operational definitions should be stated in separate section.

We used four groups of variables, namely variables of children, variables of households, variables of household heads, and variables of mothers/primary caregivers. We made three paragraphs with titles of variable groups. In each paragraph, variables are listed and definitions or categories are explained. Before starting explanation about variables, we wrote “The characteristics of the 12–23-month-old children, the characteristics of their households, and the characteristics of their household-heads and mothers/caregivers were taken from the AHS dataset.” Therefore, we think that our writing in the method section is also understandable for readers.

18. How do you check multicollinearity? Better to incorporate it in the manuscript.

Lines 199-202: We have checked multicollinearity and added the following sentences as follows, “Multivariate logistic regression analysis on Penta3 included 18 variables that showed significant differences by a Chi-square test. Multicollinearity was not found between the variables according to the variance inflation factor values for each of variables.”

19. Almost all tables are mere repetitions of text descriptions, better to avoid redundancy by choosing either of the two.

We think that the result section is a part to explain results of analysis (tables), therefore, we explained the results in tables in the section.

20. Variables such as; diarrhea, bipedal edema, vitamin A supplement and smoking of heads of households are not strongly justified for its association with vaccine coverage in the discussion section. revise it again.

No diarrhea in the last 2 weeks (P=0.005), presence of bipedal edema (P=0.019), vitamin A supplement (P<0.001), and no smoking of heads of households (P=0.017) were significantly associated with having Penta3, because the P-value was less than 0.05. Therefore, we included these variables in the discussion section.

Reviewer #3:

21. The abstract section lacks sub heading. The abstract section is lengthy.

We have included subheadings in the abstract.

22. The introduction section is also too long. It should address the tried effort to decrease the burden of the problem your country, current intervention, and any future directions. Furthermore, you should give a strong gap. Your gap is not convincing.

We revised the introduction section in order to response to the comments from Reviewer 1, therefore, the section may be long. We would like reviewers and the editor to understand how we revised according to the comments from Reviewer 1.

23. In the method section, you merged method sub-sections. You need to separate and present in clear way. Data collection method, sampling procedure, Tool.

We explained the AHS 2018 in the first paragraph of the method section. Therefore, data collection method, sampling procedure, and the tool of the AHS 2018 were explained in this paragraph. After the first paragraph in the method section, we explained study participants and variables of our study separately in each paragraph.

24. There are a lot of repetitive ideas. E.g. the study period is presented in page 3 line number 103, 115....

We have erased some sentences to avoid repetition.

25. Is your tool valid and reliable? Is it developed by the authors or adopted?

This study is a secondary analysis of the AHS data and the tool was developed and validated in the AHS but not us.

26. Check the whole document for typo errors. Be consistent with use of grammars. Statistical analyses vs statistical analysis?

We have carefully checked typo errors and revised from “analyses” to “analysis” in the three parts.

27. In the result section: you need to avoid repetition of data already stated in the table. Make it short and precise.

The result section is the part to explain the results of analysis (tables), therefore, we explained the results in tables in the section.

28. Strong recommendation: The result section should be written in short and precise way.

We have revised the result section and tried to make shorter.

29. Have you checked the model assumptions? if so report the findings.

Lines 199-202: We have checked multicollinearity and added the following sentences as follows, “Multivariate logistic regression analysis on Penta3 included 18 variables that showed significant differences by a Chi-square test. Multicollinearity was not found between the variables according to the variance inflation factor values for each of variables.”

30. In the discussion section: please check the typing errors. You need to discuss the implication of the finding.

We have checked typo errors throughout the manuscript. We discussed the implication of the findings (Pent 3 coverage and factors associate with Penta 3) in the discussion section. 

31. The declaration section seems incomplete.

We have revised “Ethics Statement” in the submission process. We will follow the guidance by the journal office if any revision is required.

---

## [Decision Letter · Decision Letter 2]

9 Jun 2023

PONE-D-22-30048R2Factors associated with pentavalent vaccine coverage among 12–23-month-old children in Afghanistan: A cross-sectional studyPLOS ONE

Dear Dr. Yamamoto,

Thank you for submitting your manuscript to PLOS ONE. After careful consideration, we feel that it has merit but does not fully meet PLOS ONE’s publication criteria as it currently stands. Therefore, we invite you to submit a revised version of the manuscript that addresses the points raised during the review process.

We look forward to receiving your revised manuscript.

Kind regards,

Orvalho Augusto, MD, MPH

Academic Editor

PLOS ONE

Additional Editor Comments (if provided):

This is the 3rd revision of this manuscript. The authors responded partially to the questions raised by the reviewer and the academic editor. At times presented unsatisfactory excuses for not taking the suggestions to improve the manuscript.

I. Particularly this question:

- Please do not use only p-values on table 1 through table 3 (it would be fine to remove these p-values). In fact, many of these p-values are repeated in the unadjusted analysis of table 4. For tables 1 to 3 do: 1) make the percentages to be by column; 2) add a column for coverage and its 95% confidence interval.

The authors state that it is a matter of style. This is a serious misconception. This is a succinct presentation of the current results in this manuscript. The reader would benefit if the row percentages was presented as coverage [the column I am suggesting to add]; and the characteristics were presented in column percentages. The p-values have no role here.

II. Also, pay attention to reviewer 3 in the last revision comments (as per his/hers recommendation). Particularly, please, respond fully to the following questions:

• The question about model assumption was not fully responded to. At least please state how did you check model fit which is closer to model assumptions check requested by the reviewer. Yes, checking multicollinearity is important but it is not enough.

• Change the term multivariate to multivariable

Reviewers' comments:

Reviewer's Responses to Questions

**Comments to the Author**

1. If the authors have adequately addressed your comments raised in a previous round of review and you feel that this manuscript is now acceptable for publication, you may indicate that here to bypass the “Comments to the Author” section, enter your conflict of interest statement in the “Confidential to Editor” section, and submit your "Accept" recommendation.

Reviewer #3: (No Response)

2. Is the manuscript technically sound, and do the data support the conclusions?

Reviewer #3: No

3. Has the statistical analysis been performed appropriately and rigorously? 

Reviewer #3: Yes

4. Have the authors made all data underlying the findings in their manuscript fully available?

Reviewer #3: Yes

5. Is the manuscript presented in an intelligible fashion and written in standard English?

Reviewer #3: Yes

6. Review Comments to the Author

Reviewer #3: Dear authors thank you for your effort in addressing the previous comments. However, all of my comments not addressed satisfactorily.

7. PLOS authors have the option to publish the peer review history of their article (what does this mean?). If published, this will include your full peer review and any attached files.

Reviewer #3: No

---

## [Author Response · Author response to Decision Letter 2]

4 Jul 2023

We would like to thank the editor and the reviewers for reviewing our manuscript. We have revised the manuscript according to your comments, which were very helpful. The revisions have been completed and the responses are as follows.

1. Please do not use only p-values on table 1 through table 3 (it would be fine to remove these p-values). In fact, many of these p-values are repeated in the unadjusted analysis of table 4. For tables 1 to 3 do: 1) make the percentages to be by column; 2) add a column for coverage and its 95% confidence interval. The authors state that it is a matter of style. This is a serious misconception. This is a succinct presentation of the current results in this manuscript. The reader would benefit if the row percentages was presented as coverage [the column I am suggesting to add]; and the characteristics were presented in column percentages. The p-values have no role here.

According to the editor’s comment, we have revised Tables 1-3 by erasing p-values by chi-square test and adding the number and percentage (and 95%CI) of children who completed Penta3.

2. Also, pay attention to reviewer 3 in the last revision comments (as per his/hers recommendation). Particularly, please, respond fully to the following questions: The question about model assumption was not fully responded to. At least please state how did you check model fit which is closer to model assumptions check requested by the reviewer. Yes, checking multicollinearity is important but it is not enough.

We have checked the model assumption by Hosmer-Lemeshow test and it showed that the force-entry method was not a good fit. Therefore, we performed multivariable stepwise logistic regression analysis with forward-selection and backward-selection and Hosmer-Lemeshow test showed that these two models were a good fit. In Table 4, we have erased the results of the force entry method and added the results of stepwise forward-selection and backward-selection methos. We have also added the results of Hosmer-Lemeshow test in the footnote of Table 4. We have revised the abstract, method, result, and discussion sections according to the change of multivariable analysis.

3. Change the term multivariate to multivariable.

We have revised “multivariate” to “multivariable” in the manuscript.

---

## [Decision Letter · Decision Letter 3]

26 Jul 2023

Factors associated with pentavalent vaccine coverage among 12–23-month-old children in Afghanistan: A cross-sectional study

PONE-D-22-30048R3

Dear Dr. Yamamoto,

We’re pleased to inform you that your manuscript has been judged scientifically suitable for publication and will be formally accepted for publication once it meets all outstanding technical requirements.

Kind regards,

Orvalho Augusto, MD, MPH

Academic Editor

PLOS ONE

Additional Editor Comments (optional):

Reviewers' comments:

Reviewer's Responses to Questions

**Comments to the Author**

1. If the authors have adequately addressed your comments raised in a previous round of review and you feel that this manuscript is now acceptable for publication, you may indicate that here to bypass the “Comments to the Author” section, enter your conflict of interest statement in the “Confidential to Editor” section, and submit your "Accept" recommendation.

Reviewer #3: All comments have been addressed

2. Is the manuscript technically sound, and do the data support the conclusions?

Reviewer #3: Yes

3. Has the statistical analysis been performed appropriately and rigorously? 

Reviewer #3: Yes

4. Have the authors made all data underlying the findings in their manuscript fully available?

Reviewer #3: Yes

5. Is the manuscript presented in an intelligible fashion and written in standard English?

Reviewer #3: Yes

6. Review Comments to the Author

Reviewer #3: All of my comments are well addressed. Now, the article is suitable for publication after careful gramer edition and proof reading.

7. PLOS authors have the option to publish the peer review history of their article (what does this mean?). If published, this will include your full peer review and any attached files.

Reviewer #3: No

---

## [Editor Report · Acceptance letter]

31 Jul 2023

PONE-D-22-30048R3 

Factors associated with pentavalent vaccine coverage among 12–23-month-old children in Afghanistan: A cross-sectional study 

Dear Dr. Yamamoto:

I'm pleased to inform you that your manuscript has been deemed suitable for publication in PLOS ONE. Congratulations! Your manuscript is now with our production department. 

Kind regards, 

on behalf of

Dr. Orvalho Augusto 

Academic Editor

PLOS ONE